

# Evaluating the influence of laser wavelength and detection stage geometry on optical detection efficiency in a single particle mass spectrometer.

Nicholas Marsden[1], Michael J. Flynn[1], Jonathan W. Taylor[1], James D. Allan[1,2], and Hugh Coe[1]

[1]Centre for Atmospheric Science, School of Earth, Atmospheric and Environmental Sciences, University of Manchester, Manchester, M13 9PL, UK
[2]National Centre for Atmospheric Science, University of Manchester, Manchester, UK

*Correspondence to:* H.Coe (hugh.coe@manchester.ac.uk)

**Abstract.** Single particle mass spectrometry (SPMS) is a useful tool for the on-line study of aerosols with the ability to measure size resolved chemical composition with a temporal resolution relevant to atmospheric processes. In SPMS, optical particle detection is used for the effective temporal alignment of an ablation laser pulse with the presence of a particle in the ion source, and gives the

option for aerodynamic sizing by measuring the offset of particle arrival times between two detection stages. The efficiency of the optical detection stage has a strong influence on the overall instrument performance.

A custom detection laser system consisting of a high powered fibre coupled Nd:YAG solid state laser with a collimated beam was implemented in the detection stage of a LAAP-TOF single particle

mass spectrometer without major modifications to instrument geometry. The use of a collimated laser beam permitted the construction of a numerical model that predicts the effects of detection laser wavelength, output power, beam focussing characteristics, light collection angle, particle size and refractive index on the active detection volume in the detection stage. We compare the model predictions with an ambient data set acquired during the Ice in Clouds Experiment - Dust (ICE-D)

project.

The new laser system resulted in an order of magnitude improvement in instrument sensitivity to spherical particles in the size range 500-800nm compared to a focussed 405nm laser diode system. The model demonstrates that the limit of detection in terms of particle size is determined by the scattering cross section ($C_{sca}$) as predicted by Mie theory. In addition, if light is collected over

a narrow collection angle, oscillations in the magnitude of $C_{sca}$ with respect to particle diameter result in a variation in the active detection volume, resulting in large particle size dependent variation in detection efficiency across the particle transmission range. This detection bias is imposed on the aerodynamic size distributions measured by the instrument and accounts for a detection bias towards sea salt particles in the ambient data set.



## 1  Introduction

Ambient measurements of aerosol properties are required to further our understanding of the role of aerosol in climate (Boucher et al., 2013) and the detrimental effects on human health (Pope and Dockery, 2006). The dynamic nature of atmospheric processes make the on-line measurement of aerosol properties very challenging and the choice of techniques depends on the specific aerosol property of interest and the temporal resolution required. In the field of atmospheric science, the importance of refractory aerosols in atmospheric processes (Formenti et al., 2011), (Hudson et al., 2011), has increased the demand for on-line measurement of size resolved composition at low number concentrations.

In single particle mass spectrometry (SPMS), high powered laser pulses are used to vaporise and ionise ambient aerosols that have been focussed into a narrow particle beam. The generated ions are usually analysed by time-of-flight mass spectrometry (TOF-MS), providing detailed composition information on a particle by particle basis. This approach allows for the direct linking of composition with physical properties such as size, shape and density (Murphy et al., 2004; Vaden et al., 2011) as well as the the probing of internal mixing state (Pratt and Prather, 2010). The type of material that can be analysed is determined by the power and wavelength of the pulsed laser system whilst the temporal resolution is related to the probability that a particle will be coincident with the laser pulse in the ion source, a phenomenon often referred to as hit-rate. In many systems, optical particle detection is incorporated into the instrument to provide an external trigger to the pulsed laser; temporally aligning the laser pulse with the presence of a particle in the ionisation region, resulting in an increased hit-rate.

The incorporation of two optical detection stages into the instrument allows the determination of particles velocity by measuring the time taken for particles to travel the known distance between two detection stages. When used with an aerodynamic lens inlet, particle velocity is a function of the vacuum aerodynamic diameter, allowing for a particle size measurement to be made in addition to the single particle composition measurement made by the TOF-MS. The design of the optical detection stage has a strong influence on the overall instrument performance and has been the focus of much research (Murphy, 2007). The features of SPMS instruments have previously been described by (Sullivan and Prather, 2005), (Nash et al., 2006), (Murphy, 2007), and (Hinz and Spengler, 2007).

Optical particle detection techniques are well established for the size resolved counting of particles in Optical Particle Counters (OPC)(McMurry, 2000), which are similar in many respects to the optical detection systems employed in SPMS. Particles are detected by collecting the scattered light generated from the interaction of a particle beam with a cw laser. When employed in SPMS, the scattering signals need only exceed a certain threshold to register a particle event, unlike standard OPCs which optically sizes the particle based on the magnitude of the scattering signal. Incorporating an optical system into the geometry of a SPMS creates additional design challenges.



Advances in laser technology have influenced the design and development of the optical detection stage in SPMS. Early instruments utilised Helium-Neon gas laser with a wavelength of $633nm$ (Hinz and Kaufmann, 1996) (Prather et al., 1994) (Murphy and Thomson, 1995) with an output in the range of $4 - 10mW$. The implementation of Nd:YAG solid state lasers $532nm$ with an output in

the range of $50 - 300mW$ greatly improved laser fluence and beam quality in later generation instrument (Thomson et al., 2000), (Su et al., 2004), (Zelenyuk and Imre, 2005), (Brands et al., 2011). Other groups have opted for newly developed laser diodes (Gaie-Levrel et al., 2012) that, while producing less output ($40mW$), have shorter wavelengths ($405nm$), and are relatively cheap and easy to implement. Shorter wavelengths are desirable when sampling particles whose diameter

(D) is smaller than the wavelength of the incident radiation as $C_{sca}$ is proportional to $D^6$ in the Rayleigh regime

The most efficient particle detection systems use an elliptical mirror to collect scattered light over a wide angle, thus maximising scattering signal at the detector (Su et al., 2004), (Zelenyuk and Imre, 2005). However, the physical dimensions of a standard elliptical mirror prevent the detection

stage being located within the ionisation region. Consequently, it must be located up-stream in the vacuum housing and complex trigger circuits must be made that produce a particle size dependent trigger delay (Zelenyuk and Imre, 2005). Such systems have excellent particle detection efficiencies, but have a hit-rate that is limited by the probability of hitting a particle with the pulsed laser.

The Laser Ablation Aerosol Particle Time-Of-Flight (LAAP-TOF) is a type of single particle mass

spectrometer manufactured by Aeromegt (GmbH) and is in an early stage of commercial development. The instrument features an aerodynamic lens inlet (model $LPL - 2.5$, Aeromegt GmbH), a bipolar TOF analyser (TOFWerks AG) and a novel particle detection system based on 405nm laser diode technology and a compact light collection optics assembly consisting of fibre optic guides that collect scattered light over a narrow scattering angle, and is located directly in the ionisation region.

We present an evaluation of the instrument performance with the original instrument manufacturer (OEM) detection stage design that identified the optical detection system as a limiting factor in instrument performance. The instrument performance is compared with a customised detection system in which the detection laser is replaced with a fibre coupled 532nm 1W Nd:YAG solid state laser system with a collimated laser beam. The influence of detection stage geometry is evaluated

using the customised detection system as the laser intensity distribution within a collimated beam is relatively simple to model because unlike a focused beam, there is no variation along the beam axis due to depth of field.

## 2 Description of Instrumentation

The instrument design has previously been described by Gemayel et al. (2016). Here, we provide a

brief overview of the instrument layout and describe the modifications made to the optical detection

system. A schematic layout of the instrument is shown in Figure 1. Aerosol enters the instrument via a $100\mu m$ critical orifice and passes through an aerodynamic lens before beam expansion into the first low pressure region of the instrument. The particle beam passes through second and third differentially pumped stages separated by skimmers that remove the majority of the gas phase. The

first detection stage is encountered in the third pumping stage when the particles pass through a cw laser beam arranged orthogonally to the axis of the particle beam. After passing a differentially pumped aperture into the TOF vacuum region, the particles encounter the second detection stage which is located within the extraction optics of the TOF analyser. The second detection stage triggers an excimer laser (ArF $\lambda = 193nm$, model EX5, GAM) to fire an intense pulse in a direction that is

co-axial, but counter-propagate with the particle beam. The cloud of ions generated by the interaction of the material with the high energy pulse are extracted into two linearly opposing TOF analysers, for positive and negative ions respectively.

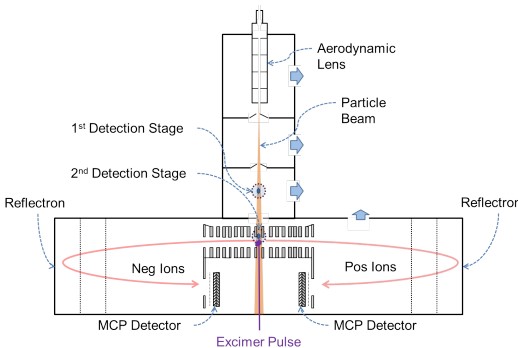

**Figure 1.** Schematic diagram of the LAAP-TOF instrument.

Instrument control is performed using IgorDAQ software (AeroMegt Gmbh) based on IGOR (Wavemetrics) which incorporates TOFDAQ (TOFWerks AG) data acquisition. The instrument can

be operated in three modes: Auto Trigger, Both Lasers, and Second Laser Only. In 'Auto Trigger' mode, the excimer is set to fire at a user defined frequency (up to 100 Hz) and mass spectra will be generated at a rate that is determined by the probability of a particle being synchronous with sufficient energy from the 8ns excimer pulse in the ionisation region. In 'Second Laser Only' mode, the excimer laser will fire when it receives a trigger signal (with a set delay) from the second detection

stage, increasing the probability of generating a mass spectra. In 'Both Laser' mode, the instrument is actively sizing particles and a trigger is required from both the first and second detection stages in order to fire the excimer laser.

The first and second detection stages have very similar designs. A high powered cw laser beam, arranged in a vertical orientation, passes through the instrument to a beam dump on the underside

of the vacuum chamber. The instrument is aligned so that the particles in the particle beam interact





with the detection laser radiation in a position where scattered light can be collected by a set of 12 optical fibres, arranged in a concentric pattern centred on the axis of the cw laser beam. The diameter of the circle defined by the naked fibre termini, and the position below the axis of the particle beam is such that light is collected over an angle of $\sim 13 - 15°$ with respect to the incident laser radiation

at the first detection stage, and $\sim 10 - 12°$ with respect to the incident laser radiation at the second detection stage. The compact design of the collection optics allows the detection of particles within the ionisation region of the mass spectrometer, negating the requirement for size dependent trigger delays that are a feature of some systems.

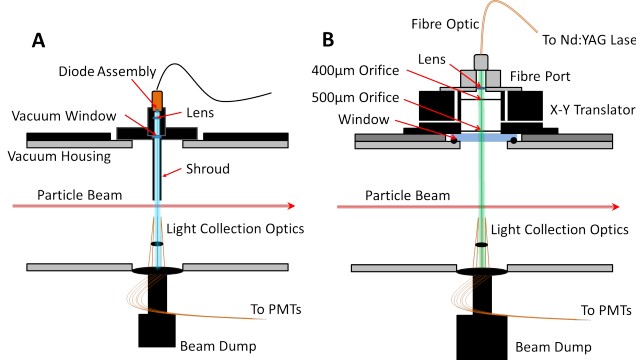

**Figure 2.** Schematic diagrams of the LAAP-TOF detection stages. System (A), a 405nm diode based system with a focussed beam ($D4\sigma$ of $50\mu m$) and a sheath type spatial filter. System (B), a custom built 532nm Nd:YAG DPSS fibre coupled system producing a collimated beam with a $1/e^2$ waist diameter of $330\mu m$, and spatial filtering outside the vacuum housing.

The optical fibres for light collection are connected to 2 photomultiplier tubes (PMT) so that each

PMT receives light from 6 fibres. A PMT signal pulse height above a user set threshold results in a transistor-transistor logic (TTL) pulse from the discriminator to the microprocessor control unit (MCU) in the timing electronics. A pulse is required from both PMTs in the detections stage (1 and 3 for the first detection stage, 2 and 4 for the second detection stage) in order for the MCU to recognise the presence of a particle. The thresholds are set experimentally by finding the minimum

threshold value at which false triggers (noise) are not created; resulting in the discrimination between high frequency noise and true particles, improving the signal to noise performance of the detection system in a similar set-up as described by Trimborn et al. (2000).

The delivery of a high powered cw laser beam into the optical detection region requires a system with three parts; a laser device that outputs a high powered coherent beam at a wavelength that

is dependent on the lasing media, an optical assembly that incorporates optical focussing and beam steering capability, and spatial filtering that cleans-up the beam profile in order to reduce background noise on the detection system. In this study, two different systems of laser delivery are evaluated;





Detection System A, based on a laser diode emitting light at 405nm that is focussed by a lens with a long focal point, and spatially filtered with a sheath within the vacuum chamber; and Detection

System B, based on a custom built fibre coupled system featuring a Nd:YAG diode pumped solid state laser, emitting at 532nm that is collimated by a fiberport and spatially filtered by an orifice outside the vacuum chamber.

In System A, a compact 450mW opnext diode module (HL40023MG, Thorlabs) is mounted in a threaded copper tube and fitted to an adjuster assembly to produce a cw laser beam of 405nm

wavelength orthogonal to the particle beam axis (Figure 2 A). The adjuster assembly incorporates a focussing lens to create a beam with a $D4\sigma$ (second moment width) focal point diameter of $51.2\mu m$ measured with a beam profiler at a focal length of $50mm$. Power output of the system can be adjusted by varying the drive to the diode module. The position of the focal point relative to the particle beam can be adjusted by moving the whole adjuster assembly relative to the vacuum housing in the

horizontal X and Y directions and varying the focal length in Z direction.

System B comprises of a 1W GEM 532nm Nd:YAG laser (Laser Quantum Ltd) with divergence $< 0.8mrads$, pointing stability $< 10\mu rad/^0C$ and a beam quality factor $(M^2) < 1.1$ enabling the beam to be efficiently launched into a single mode fibre and focussed into a collimated beam with just one lens. An aspheric fiberport (model PAF-X-2A, Thorlabs) with an output $1/e^2$ waist diameter

of $330\mu m$ and a divergence of 1.75mrads was chosen to collimate the beam. A maximum waist distance of 96mm ensured the beam remained collimated at the light collection optics therefore reducing background noise. The fiberport lens has an input Mode Field Diameter (MFD) of $3.5\mu m$.

The adjustable fiberport allows for precise positioning of the lens for efficient collimation and allows the beam to be steered through the spatial filters (Figure 2B). Spatial filtering is required to

clean up the beam profile and reduce background noise. A $500\mu m$ orifice is placed at the bottom of the optical stack, and the collimated beam can be accurately positioned through the pin hole using the steering on the fiberport. Light scattering from the pin hole is reduced by blackening the metallic orifice with Aquadag and introducing an intermediate $400\mu m$ orifice that is aligned to the beam axis beam using an X-Y translator. The optical stack can be aligned on the bench before being fitted to

the instrument and aligned to the lowest noise position. A N-BK7 coated laser window (Thorlabs) was used to seal the vacuum.

## 3 Methods

A number of methodologies were used to evaluate the instrument performance. The creation of aerosols in the laboratory was required for the instrument set-up procedure, instrument sensitivity

testing and the subsequent evaluation of performance of the particle detection system. Ambient sampling was carried out to measure the sensitivity of the instrument in atmospheric conditions and



assess the suitability of the system for in-situ measurement of size resolved chemical composition of atmospheric aerosols.

### 3.1 Laboratory Experiments

Monodisperse particles were produced using a ATM226, (Topas GmbH) polystyrene latex spheres (PSL) (Duke Scientific, Inc) that were subsequently dried and passed to a custom built Scanning Mobility Particle Sizer (SMPS). The SMPS comprised of a Differential Mobility Analyser (DMA) (3081, TSI, Inc) and Condensation Particle Counter (CPC) (3786-LP, TSI, Inc). The flow was divided between the CPC and LAAP-TOF after the DMA using an equal 'Y' splitter and conductive tubing.

A measurement of the particle beam density profile was made using the detection laser to probe the particle density in a experiment similar to the optical detection method described by Petrucci et al. (2000). The front and rear optical detection systems were aligned onto the particle beam which was placed in a central position with respect to the instrument axis. The particle beam was then traversed in the horizontal axis (orthogonal to the detection beam axis) using the lens adjuster to

pivot the inlet in the gimbal in 0.05mm intervals. The number of particle pulses was measured over a 1 minute period at each position using an oscilloscope. The lateral movement of the particle beam in the plane of detection can be easily calculated with simple trigonometry. A particle density profile was then constructed using the data.

### 3.2 Field Deployment

The instrument was deployed for ground-based ambient aerosol characterisation during the Ice in Clouds Experiment (ICE-D) project in August 2015. ICE-D was a multi-platform field campaign with the primary aims of studying aerosol-cloud interactions during the evolution of towering cumulus clouds over the sub-tropical Atlantic Ocean. The project had significant aircraft and ground operations that involved the in-situ characterisation of the properties of mineral dust advected from

the Sahara Region.

Ground-based aerosol characterisation took place at Praia International Airport, Santiago Island, Cape Verde from 27th July to 23 August 2015. The ground site comprised of the Manchester Aerosol Container with a 30m scaffold tower for ambient aerosol sampling. The tower supported a pumped inlet that was divided into heated lines inside the aerosol container, delivering ambient aerosol to a

host of instruments including; the LAAP-TOF and an Aerodynamic Particle Sizer (APS)(3321, TSI Inc). The APS measures aerodynamic particle diameters in the range $0.5 - 20\mu m$ which provided a suitable comparison to the sampling efficiency of the LAAP-TOF.

Previous studies at this location have recorded a multi-modal aerosol size distribution with number concentrations up to 100/cc in the size range $0.5 - 2.5\mu m$ during dust events (Kandler et al.,

2011). Taking into account particle loses in the sampling lines, a sampling efficiency of $0.01 - 0.1$



was required for the LAAP-TOF during the ICE-D campaign in order to measure sized resolved composition at a temporal resolution of several particles per minute.

### 3.3 Instrument Performance Evaluation

To characterise the instrument performance we describe the elements that contribute to the efficiency of the instrument in each acquisition mode. An aerodynamic lens focuses particles of different size and shape with different efficiency (Liu et al., 1995), (Zhang et al., 2004). The transmission efficiency of the aerodynamic lens ($E_{Lens}^d$) is defined as the proportion of spherical particles with physical diameter $d$ that exit the expansion nozzle relative to the number of particles that passed the critical orifice i.e. measure of the losses that occur within the lens assembly and critical orifice holder.

On supersonic expansion into the low vacuum of the instrument, a particle beam is formed whose solid angle of divergence relative to the lens axis defines a cross-sectional area at a set distance from the nozzle expansion. This area has a 2D Gaussian probability density function (2DG-PDF) (Huffman et al., 2005). The probability of hitting a particle and obtaining a mass spectrum within this 2DG-PDF is known as the Hit-Rate efficiency ($E_{Hit}^d$), and is the product of the geometrical overlap of the excimer laser focal point with the particle beam ($E_{Geom}^d$), the temporal overlap of the peak UV laser power with the presence of a particle in ionisation region ($E_{Temp}^d$), and the ionisation efficiency of the UV laser ($E_{Ion}^d$) with respect to the particle composition (Erdmann et al., 2005).

The overall efficiency of the instrument to generate mass spectrum from an ambient aerosol population is acquisition mode specific. In the case of 'Auto Triggering' mode, the probability that a mass spectrum will be generated ($E_{AutoMS}^d$) is simply the product of $E_{Lens}^d$ and $E_{Hit}^d$. The probability of hitting a particle and obtaining a mass spectrum in 'second laser only' mode includes an extra term relating to the size dependent efficiency of optical detection ($E_{Detect}^d$). The hit-rate is also modified with respect to free firing mode because the term $E_{Temp}^d$ is modified to a probability relating to the trigger delay and the size dependent particle velocity. In addition, the active area of the UV ionisation is assumed to be larger than the active area of detection, so that $E_{Geom}^d = 1$ in this acquisition mode, providing the instrument is correctly aligned. We defined the overall efficiency of 'second laser only' mode as $E_{TriggeredMS}^d$, and using the definitions above is defined by Eq. (1). Note that all terms have a particle size dependence.

$$E_{TriggeredMS}^d = E_{Lens}^d * E_{Detect}^d * E_{Hit}^d \tag{1}$$

The third acquisition mode, referred to as 'both lasers' in the software, involves the particle detection at two distinct stages so that the aerodynamic size of a particle can be measurement before the particle is ablated. The two detection stage will have different detection efficiencies because of slightly different geometries and different particle beam widths related to the down-stream distance



from the nozzle of the aerodynamic lens. We defined the sampling efficiency in 'both lasers' mode as $E_{SizedMS}$.

### 3.4 Modelling the optical detection geometry

The portion of the divergent particle beam sampled by the orthogonally incident detection laser defines an active area of detection $A_{detect}$, where the intensity of the radiation exceeds a minimum

threshold ($I_{min}$) to produce enough scattered light from the interaction of radiation with the particle for the detection system to register a particle event. The resulting efficiency of optical detection $E_{Detect}$ is a function of $A_{detect}$ and and the particle number density of the portion of the $2DG-PDF$ particle beam that it covers, which have a size and shape dependence. For simplicity, we modelled spherical particles, so the size dependence only was considered.

It has been shown that a 1D Gaussian model is sufficient to quantify the amount of a particle beam blocked by a thin wire (Jayne et al., 2000), (Huffman et al., 2005). In the case of optical particle detection by a collimated detection laser, we assumed that the detection beam is perfectly centred on the particle beam, so that the portion of the particle beam that was actively detected is the portion covered by an active radius of detection ($R$) in the radial direction ($r$) from the centre of the

particle beam. The general relationship between particle beam width $\sigma_p$ and $R$ in Eq. (2) was used to quantify $E_{Detect}$.

$$E_{Detect}^d = \frac{2}{\sqrt{\pi}} \int_0^R e^{R^2/2\sigma_p^2} dr = erf\left(\frac{R}{\sqrt{2}\sigma_p}\right) \tag{2}$$

The relationship between $R$ and $E_{Detect}$ is demonstrated in figure 3 for several nominal particle beam widths. An $E_{Detect}$ of close to 1 is achieved when the detection width R is equivalent to $3\sigma_p$

width. Note that when the $R < \sigma_p$, doubling the $\sigma_p$ has the effect of halving the detection efficiency. This loss of detection efficiency with $\sigma_p$ diminishes when $R > \sigma_p$.

The laser beam power density profile is described by a 2DG-PDF orthogonal to the laser beam axis. As the particle beam is orthogonal to the laser beam axis, a 1D Gaussian PDF describes the laser beam intensity profile that is encountered by particles as they cross the detection stage. The

active radius of detection $R$ was modelled for a specific detection laser width $\sigma_{Detect}$ and power $a$ by the relationship in Eq. (3).

$$R = \sigma_{Detect} \sqrt{-ln\left(\frac{I_{min}}{a}\right)*2} \tag{3}$$

The effects of the relationship shown in equation3 are two fold. Firstly, the choice of detection laser beam waist size ($\sigma_{Detect}$) determines the active area of detection at a set $I_{min}$. Secondly,

$I_{min}$ has a controlling influence on the portion of the collimated beam that will be active. Figure 4 demonstrates the relationship of $R$ and $\sigma_{Detect}$ for nominal values of $I_{min}$. In these examples, the





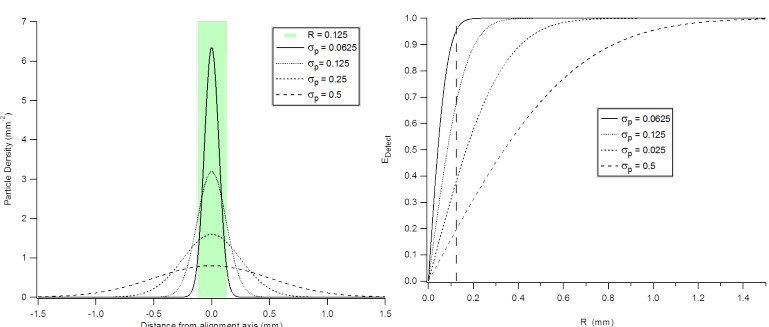

**Figure 3.** (a)The overlap of $R = 0.125mm$ with four normalised particle beams with widths $\sigma_p = 0.0625, \sigma_p = 0.125, \sigma_p = 0.25, \sigma_p = 0.5$. (b) The relationship of the active detection width $R$ and $E_{Detect}$ for a range of $\sigma_p$ . $R = 0.125mm$ is marked for reference.

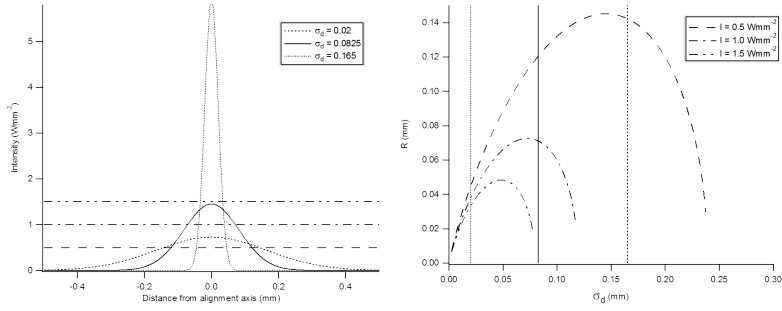

**Figure 4.** (a) Gaussian intensity distribution for collimated laser beams of fixed power focussed to three different beam waist diameters. (b) The relationship of the active detection radius (R) with the beam width ($\sigma_{detect}$) for $I_{min}$ of 0.5, 1.0 and 1.5 $Wmm^{-2}$. The beam widths shown in the left panel are marked for reference

output power of the laser is constant, but collimated to a range of $\sigma_{Detect}$ beam widths. For small detection beam widths, there is a weak dependence on $I_{min}$ as most of the density profile exceeds the threshold (Figure 4A). However, at wider particle beams, larger values of R are produced at the expense of greater $I_{min}$ dependent variability. For example, with a detection beam width of $\sigma = 0.10mm$, the active beam width at an intensity threshold of $0.5Wmm^{-2}$ is double that of an intensity threshold of $1.0Wmm^{-2}$, but will fail to detect particles with an $I_{min} > 1.25Wmm^{-2}$ as the active width is $0mm$.

The effective width of the detection laser is smaller than the $4\sigma D$ beam width as only the portion of the profile that exceeds $I_{min}$ is active. The magnitude of $I_{min}$ is a function of the scattering cross section of the particle $C_{Sca}$, and a transfer function $K$ that describes the minimum amount of power transfer to the detector required, and accounts for the collection and transfer of light by the fibre optics, the radiant sensitivity of the detector, and the characteristics of the electronics in the



trigger unit. The relationship between $I_{min}$, $K$ and $C_{Sca}$ is shown in Eq. (4). The transfer function
is assumed to be the same for all particles (Jonsson et al., 1995) while $C_{Sca}$ has a strong dependence
on the size and refractive index of the material.

$$I_{min} = \frac{K}{C_{Sca}} \tag{4}$$

For a given wavelength of incident light, Mie theory provides an exact solution to the scattering of
light by a sphere of known size and refractive index. The scattnlay algorithm (Peña and Pal, 2009)
was used to model $C_{Sca}$ for a variety of collection angles, particle sizes and incident wavelengths.
Examples of how $C_{Sca}$ varies with particle size and wavelengths are demonstrated in figure 5.In
these examples, the light collection is set $10-12°$, representing the rear detection optics assembly
and the refractive index is 1.59. The wavelengths of commercially available laser systems are plotted.

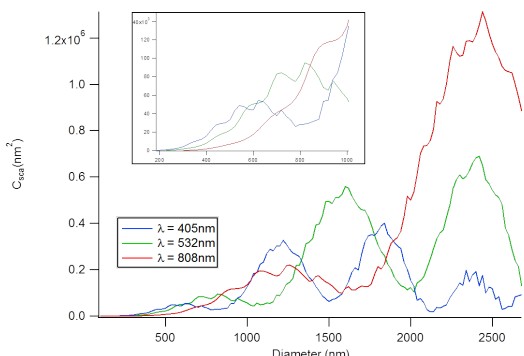

**Figure 5.** $C_{sca}$ versus particle diameter modelled using the scattnlay algorithm for wavelengths used in commercially available lasers. $C_{sca}$ for particle diameters $200-1000nm$ are shown in more details in the inset. Refractive index = 1.59, collection angle $10-12°$.

## 4 Results

The efficiency with which the instrument can sample an aerosol population determines the temporal resolution in which data can be collected from a specific number size distribution. This makes the mode specific sampling efficiency an important parameter in SPMS. Results of measuring the sampling efficiency in 'second only' mode with detection laser system A and system B are reported below followed by an analysis with the optical detection model.

### 4.1 Instrument Performance


In this study, it was convenient to assess the instrument performance by measuring $E^d_{TriggeredMS}$ experimentally, as this mode involves the evaluation of the hit-rate efficiency using only one detec-



tion stage. We directly measured $E^d_{TriggeredMS}$ in the laboratory by comparing size selected number
concentrations measured with a CPC, with particle counts observed with the LAAP-TOF. Results of
this measurement using detection system A are shown in Figure 6 A. The data shows a characteristic
steep drop-off in spectral efficiency for small particles described by others for instruments operating
in a similar acquisition mode (Cziczo et al., 2003), (Cziczo et al., 2006). The minimum sized parti-
cles that could be detected with the 405nm diode system were 350nm and the maximum efficiency
achieved was a little over $0.01$ for $E^{600nm}_{TriggeredMS}$. For particle $> 600nm$, the efficiency decreases
with increasing particle size so that $E^{800nm}_{TriggeredMS} = 0.005$. This result is in good agreement with
LAAP-TOF detection efficiency data recently reported by (Gemayel et al., 2016) who detected a
minimum particle size of 350nm, a peak detection efficiency of 0.025 at 450nm and decrease in
sensitivity to particle with diameters of 800nm.

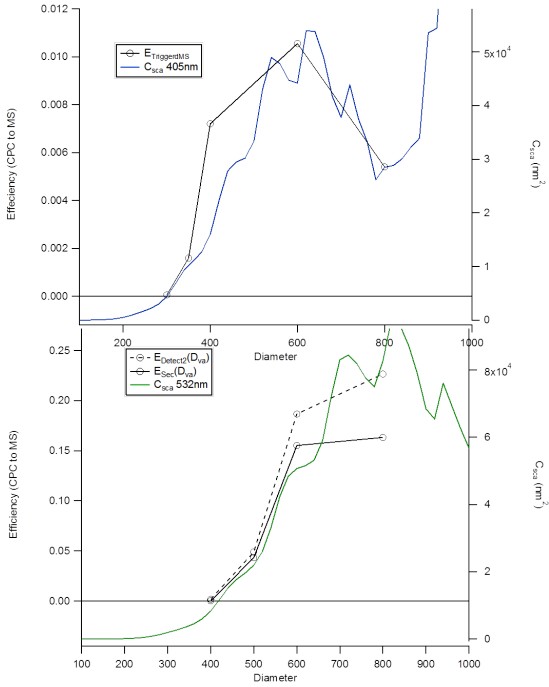

**Figure 6.** Spectral acquisition efficiency in 'second only' acquisition mode ($E^d_{TriggeredMS}$) using detection
system A (top panel) and detection System B (bottom panel). Measurements are the comparison of number
concentration obtained by the LAAP-TOF with a CPC when sampling size selected laboratory generated PSL
aerosol of various diameter.

To analyse the contributing factors to the reported efficiency we examined the data for 600nm
spherical particles in more detail. It is well established that an aerodynamic lens can achieve an ef-
ficiency close to unity for the size range under examination here $(200 - 600nm)$ (Huffman et al.,



2005), (Williams et al., 2013) therefore, although the lens is uncharacterised, we assumed $E_L^{600nm} = 1$. It was observed that the vast majority of PSL particles that produced an optical trigger also produced a mass spectrum so that $E_{Hit}^{600nm} \approx 1$. With reference to Eq. (1), we concluded that as a first ap-

proximation the observed performance was limited by the optical detection stage i.e. $E_{TriggeredMS}^{600nm} \approx E_{Detect}^{600nm}$.

$C_{sca}$ for 405nm light is plotted against particle diameter for comparison with $E_{TriggerdMS}$ in figure 6A. At the small particle sizes (200-600nm), there is a correlation between the two curves indicating that the low size cut-off in the sampling efficiency, in this case 350nm, is limited by

the scattering intensity of small particles. However, it is also clear that $C_{sca}$ function continues to oscillate as the diameter increases, and will also affect the detection efficiency of larger particles, particularly in the region 700-900nm as suggested by Gaie-Levrel et al. (2012) and Gemayel et al. (2016).

$C_{sca}$ of small particles (200-800nm) at 532nm also correlates with the sampling efficiency curve

measured with System B (figure 6B). This method of detection laser delivery results in values of $E_{TriggeredMS}$ of an order of magnitude higher than system A for particles larger than 500nm. There is evidence that $E_{Hit} < 1$ with this detection geometry. For example with $E_{TriggeredMS}^{600nm} = 0.15$ and the corresponding $E_{Detect}^{600nm} = 0.19$, the difference between the two values is due to $E_{Geom}^{600nm} < 1$, resulting in $E_{Hit}^{600nm} = 0.79$. The shift in the $E_{TriggeredMS}$ function on the two plots when changing

laser wavelength confirms that a more general case $E_{Triggered}^d \approx E_{Detect}^d$ for Detection System A and that the observed $E_{Detect}^{600nm} > E_{Detect}^{800nm}$ is an optical detection effect and not a lens transmission effect.

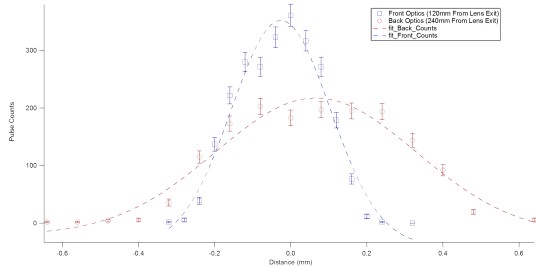

**Figure 7.** Particle beam profiles obtained using the optical characterisation method. Measurement were made using a beam of 600nm PSL at the front and rear detection stages at a distance of 115mm and 230mm from the aerodynamic lens nozzle respectively. Error bars are $1\sigma$ standard errors assuming poissons statistics.

Further analysis of $E_{Detect}^{600nm}$ required information on the particle beam width. Measurements were made using the optical characterisation of a particle beam created from the nebulisation of 600nm

sized PSLs. The resulting in particle density profile at the front and back optical detection stages are shown in Figure7. Assuming the active area of detection is very small compared to the beam



cross-sectional area, and that focal point passes through the particle beam axis, a 1D Gaussian curve fit is a sufficient approximation (Jayne et al., 2000). The beam width is estimated to be $\sigma = 0.13mm$ at detection stage 1(120mm from the lens exit), and $\sigma = 0.26mm$ at detection stage 2 (240mm from

the lens exit).

### 4.2 Optical Detection

A collimated detection laser beam, whose full $1/e^2$ width (0.330mm) exceeds $I_{min}$, transecting a $\sigma_p = 0.26mm$ wide particle beam profile would theoretically give $E_{Detect}^{600nm} = 0.53$ (Eq. (2)). This value is significantly higher than the measured value of 0.19. The most likely explanation is that

with 600nm sized particles, $R$ is significantly less than the $1/e^2$ beam diameter because the intensity threshold of detection is only exceeded by a portion of the Gaussian beam profile.

We can use the model of the optical detection geometry to calculate the effective detection beam width across the transmission range with spherical particles. Using Eq. (2) and the empirically derived $E_{Detect}^{600nm} = 0.19$, a value of $R_{detect}^{600} = 0.057mm$ is calculated, which corresponds to laser beam

intensity of $1.1Wmm^2$ for $I_{min}$. Using Eq. (4), and a $C_{sca}$ value of $5.066*10^{-8}mm^{-2}$ derived from Mie theory model, the value of the Transfer Function K is $5.59*10^{-8}W$. This allows a more general $R$ to be plotted as a function particle size using Eq. (3).

Results of modelling Detection System B with PSL particles across the theoretical transmission range of the aerodynamic lens are represented in Figure8. The data predicts a rapid drop-off in R

when the particle diameter is less than $\approx 550nm$. Mie oscillations impact R across the transmission range with distinct minima observed at $1\mu m$ and $2\mu m$ particle diameter. The model allows for the evaluation of the effective detection radius (R) when changing the optical detection stage design parameters. Figure8 A shows the effect of changing the detection laser wavelength. Shorter wavelengths produce deeper oscillations at a higher frequency than longer wavelengths. A wavelength

of 808nm produces the most stable profile but comes at the expense of the small particle detection which drops-off at 700nm.

The effect of light collection angle was modelled by changing the input parameters to the scattnlay algorithm. Light collected over $10 - 170°$ produces a smoother profile with a lower particle size cut-off than a the narrow angles represented by LAAP-TOF detection system (Figure 8 B). This is

because collecting more light increases the integrated phase function and requires less intensity in the detection beam in order to exceed $I_{min}$. The variation in R stabilises if the value of R is close to the $3\sigma_p$ detection beam width. Note that light collection at the front detection optics, represented in Figure 8B at an angle of $13 - 15°$, predicts blind spots in the transmission range. This is not thought to be the actual case for the system as the front optics have a lower background noise than the rear

optics and so will have a lower value of the Transfer Function, producing smaller amplitude variation in the profile.



Adjusting the value of the Transfer Function in the model allows for the simulation of the impact of signal-to-noise ratio on R. Increasing the signal-to-noise ratio produces a larger R over the transmission range and reduces the low particle cut-off size. However, it does not reduce the amplitude

of oscillation as much as changing the collection angle (Figure 8 C).

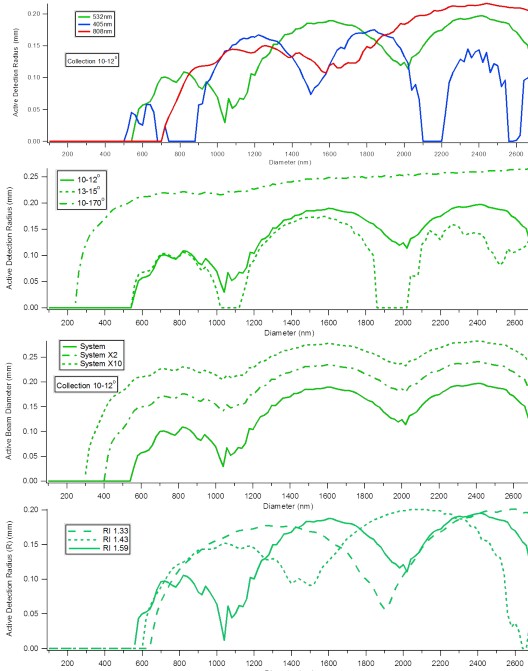

**Figure 8.** Model results showing the calculated effective detection beam radius (R) for a number of design configurations. Results of modelling R for Detection System B with PSL particles are shown in each plot for reference (using 300mW of 532nm light collimated to a beam waist of $\sigma_d = 0.0825mm$, a light collection angle of $10 - 12°$ and refractive index of 1.59). Comparative profiles are modelled by changing only one parameter with respect to the design of Detection System B. (a) The effect of changing the detection laser wavelength, (b) the effect of changing light collection angle, (c) the effect of signal-to noise and D) the effect of particle refractive index.

The choice of lens determines the width of the beam waist of the collimated beam. With a set laser power output, increasing the beam waist width has the effect of reducing the intensity at a given radial distance from the beam axis. Results of modelling of $I_{min}$ with respect to $\sigma_d$ shown in Figure 4 demonstrates that a common maximum R does not exist for variable $I_{min}$. This indicates a particle

size dependence in the optimal $\sigma_d$ value with this detection geometry. This effect is quantified in Figure 11 which shows the variation in R as a function of $\sigma_d$ and particle diameter. Small focal points give the least variation with respect to particle size but produce a relatively small effective





beam radius (R<0.1mm). Values of R can be produced that exceed 0.6mm but come at the expense
of very high variation with respect to particle size that causes blind areas in the detection profile.

**4.3 Ambient Measurements**

Size resolved composition measurement of over 250000 particles were obtained during ambient
sampling in a remote marine environment over a 14 day period . Analysis using the fuzzy c-means
clustering built-in to the LAAP-TOF Data Analysis tool (Aeromegt GmbH) placed the particles
into three main classes, a dominant NaCl class, a silicate mineral class and a minor contribution of
secondary material. A 7 day time series of the clustered particle number concentrations is shown in
Figure 9 alongside number concentration reported by the APS system. The LAAP-TOF reports total
number concentrations two order of magnitude lower than the APS. The time series trends are in
generally well correlated over the first 4 days, with divergence occurring after the arrival of silicate
dust from the Sahara Desert around the 11th August 2015. A histogram of the aerodynamic particle
size measurements of each cluster are displayed in Figure 10A. The NaCl class is reported in a mode
at 1400nm, and the silicate and secondary material are multi-modal centred on 800nm and 1400nm.

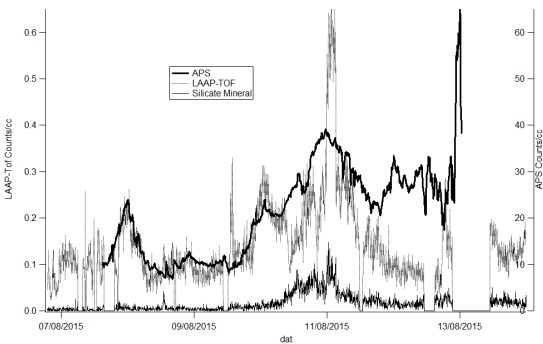

**Figure 9.** Time series of total number concentrations measured by LAAP-TOF and APS for a 7 day period
during the ICE-D campaign. Also displayed are the number concentrations of particles in the silicate mineral
class as determined by fuzzy c-means clustering.

**5 Discussion**

Optical detection of particles in SPMS has been the focus of much research (Murphy, 2007). Early
instruments used fibre optics to guide scattered light to photomultiplier tubes (PMT) for detection
(Prather et al., 1994), (Hinz and Kaufmann, 1996). The introduction of an ion source extraction plate
fashioned into an elliptical mirror (Murphy and Thomson, 1995) allowed light to be collected over a
wider solid angle within the ion source whilst maintaining relatively high hit-rates, but this approach
is not amenable to bipolar TOF due to extraction field distortion. The removal of the detection optics





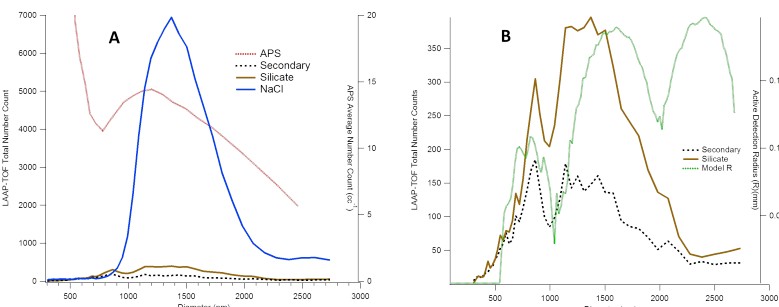

**Figure 10.** Particle size distributions of ICE-D ambient particles measured by LAAP-TOF. (a)The aerodynamic diameter distribution on the NaCl, Secondary and silicate mineral classes compared to the size distribution measured by the APS. (b) The secondary and silicate classes compared to the modelled active beam diameter of the second optical detection stage (RI 1.59, collection angle $10 - 12°$).

from the ion source region permitted a more complete elliptical mirror to be used (Gard et al., 1997),
but required a more complex trigger system to ensure the UV ablation laser hits particles travelling
a different velocities.

The aim of using elliptical mirrors is to collect as much scattered light as practicable over a wide
solid angle. The benefit of this approach is demonstrated in the model results that calculates R to be
close to the $3\sigma_d$ width across much of the transmission range. Authors have reported $E_{Detect}$ values
of 0.5 - 1 for instruments using this type of collection optics (Su et al., 2004),(Zelenyuk and Imre,
2005), (Zelenyuk et al., 2009), (Brands et al., 2011) over selected size ranges, typically $200-600nm$.
Hit-rates are generally much lower with instruments that do not trigger in the ion source and overall
sampling efficiency is rarely reported in the literature.

An ideal instrument design would have optical detection in the ion source of a bipolar TOF while
maintaining the optical detection efficiency of an elliptical mirror. This is very difficult to achieve in
practice. The compact design of the light collection optics in the LAAP-TOF allows for the detection
stage to be placed in ion source, but collects light over a very narrow angle. Our initial laboratory
studies with Detection System A showed that a low detection efficiency in the size range $300 -
800nm$ produced a sampling efficiency that was insufficient for the targeted application. However,
the hit-rate with this set-up was exceptionally high (close to 1) presumably due to the very small
detection laser focal point resulting in a high $E_{Geom}$.

Laser diode light sources have relatively low power and produce a relatively divergent beam that
can be difficult focus with a single lens. The implementation of a 532 nm Nd:YAG laser with high
pointing stability allowed for single mode fibre coupling and efficient collimation to a larger beam
waist diameter compared to the focal point of Detection System A. The implementation of Detec-
tion System B resulted in over an order of magnitude improvement in $E_{TriggeredMS}$ compared to
Detection System A, but came at the expense of a slightly larger particle size cut-off. Larger beam





diameters have the potential to increase the detection efficiency for some particle sizes. However, modelling of the effective beam diameter shows that for a set laser power output, the optimum de-
tection laser beam width is particle size dependent (Figure 11A). The particle size distribution of the target application must be considered when choosing the output power and focussing characteristics of the detection laser system if using light collection optics with a narrow collection angle.

Ambient measurements demonstrate the impact that this effect may have on the data. The sampling efficiency is low compared to the laboratory studies, probably due to the effects of particle
shape on the particle beam divergence and the low $E_{ion}$ of some of the silicate mineral dust. A Size dependent detection bias may also influence the sampling efficiency. The measurement of NaCl particle is more efficient because the size mode coincides with a maxima in R at around 1500nm. Comparison with the size distribution with the APS suggests that the size distributions of the silicate and secondary classes represent the tail of an accumulation mode with a center below 300nm, with a
size dependent detection bias imposed on the profile. The role of the particle refractive index appears to influence in the detection bias. Size distribution profiles of the silicate and secondary material have maxima that correlate with the modelled R value profile of material with a refractive index of 1.59 (Figure 10B). The size distribution of the NaCl class shows strong similarity with the R profile using a refractive index of 1.33 (Figure 8D). The light scattering properties of marine aerosol is known to
vary considerably with composition and humidity (Tang et al., 1997). Some caution must be used when interpreting particle size distributions acquired with this detection system.

Optical particle detection is an established technique in instruments dedicated to measuring particle size distributions of ambient aerosol populations (Baumgardner et al., 2011). The smoothing of Mie scattering undulation is a design requirement when accurate particle size measurement requires
a near monotonical response in pulse magnitude with respect to particle diameter (McMurry, 2000). Instruments utilising monochromatic laser source require a wide collection angle, whereas a monotonical response has been reported for a near-forward scattering instrument using an incandescent (white light) source (Heim et al., 2008).

A detection laser consisting of mixed wavelengths may be beneficial to the near-forward light
collection system utilised in LAAP-TOF . The Mie theory model (Figure 5) indicates that 532nm may complement 808nm by covering the deep oscillation in the profile. The mixing of light from two distinct sources is possible with a fibre coupled system. Figure 11B shows an example of R width modelling using equal powered 532nm and 808nm wavelength sources whilst maintaining the signal-to-noise characteristics. This model shows less variation in R with respect to particle size and
offers the possible of using larger $\sigma_d$ which would improve the overall sampling efficiency.





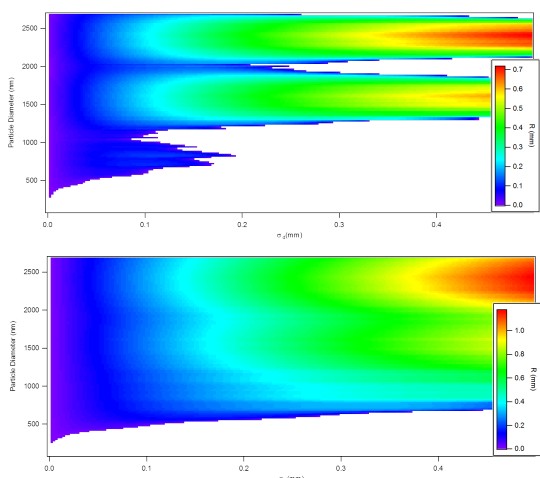

**Figure 11.** Model of the effective detection beam radius as a function of particle size and collimated beam waist size. (a) A detection beam of 300mW and a wavelength of 532nm, light collection $10-12°$ and (b) A detection beam comprising of equally mixed 532nm and 808nm laser sources.

## 6 Conclusions

A custom detection laser system consisting of a high powered fibre coupled Nd:YAG solid state laser with a collimated beam was implemented in a LAAP-TOF single particle mass spectrometer without major modifications to instrument geometry. The new laser system resulted in an order of
magnitude improvement in sensitivity to spherical particles in the size range 500-800nm compared to a focussed 405nm laser diode system. A collimated beam maximises the active area of detection whilst minimising the stray light divergence at the light collection optics, but comes at the expense of slightly reduced hit-rates compared to a focussed beam system. A numerical model is presented that allows for a general evaluation of how beam width, wavelength and light collection geometry
affect the particle detection efficiency of the optical detection stage. We used the model to explain number counting bias in an ambient data set.

    The laser intensity encountered by a particle in a collimated laser beam is a function of it s position within the Gaussian intensity distribution with respect to the laser beam axis. A calculated transfer function is used to quantify the minimum intensity requirement which defines an active detection
radius (R) that is a function of the scattering cross-section ($C_{sca}$) of the particle. The model predicts that $C_{sca}$ controls the limit of the detection in terms of particle size as expected. However, if light is collected over a narrow collection angle, Mie interference patterns result in an oscillation of $R$ with respect to particle size across the transmission range ($0.2-2.5\mu m$), resulting in large particle size dependent variation in detection efficiency. We compare the model prediction with an ambient



data set acquired during the Ice in Clouds Experiment - Dust (ICE-D) project, a multi-platform field
campaign based at the Cape-Verde Islands in August 2015. The model is used to explain a detection
bias towards sea salt particles with an aerodynamic size mode of $\approx 1.5\mu m$. We also show that the
detection bias imposes itself on the measured aerodynamic size distribution as determined by the
instrument, an effect that must be considered when interpreting the data.

Modelling of the effective detection beam diameter shows that for a set laser power output, the
optimum detection laser beam waist, or focal point size, is particle size dependent. The particle
size distribution of the target application must be considered when choosing the output power and
focussing characteristics of the detection laser system if using light collection optics with a narrow
collection angle. Variations in the active detection radius could be minimised by collecting light

over a wider angle or by mixing laser wavelengths. The stabilisation of R with respect to particle
diameter would result in more accurate aerodynamic size distribution measurements and reduce
the variation in particle number concentration measurements of different particle size, shape and
refractive index. A more rigorous evaluation of the effective of particle size and morphology on the
overall sampling efficiency $E_{SizedMS}$ would require a model of the aerodynamic lens characteristics

in order to constrain particle beam divergence. The effect of particle beam divergence on both the
variation and absolute $E_{SizedMS}$ could be improved by reducing the length of the particle flight path
by shortening the analyser housing.





## Appendix A: List of Symbols

| Symbol | Description |
| --- | --- |
| $d$ | Particle diameter. |
| $\sigma_p$ | Particle beam width. |
| $\sigma_d$ | Detection beam width. |
| $C_{sca}$ | Scattering cross section. |
| $R$ | Active radius of Detection. |
| $I_{min}$ | Minimum intensity threshold required to produce enough scattered light to register a particle event. |
| $K$ | Transfer function of the optical detection stage. |
| $E_{Lens}$ | Transmission efficiency of the aerodynamic lens. |
| $E_{Hit}$ | Hit rate efficiency. |
| $E_{Detect}$ | The particle detection efficiency of the optical detection stage. |
| $E_{Geom}$ | Geometric overlap of the excimer laser pulse with the particle beam. |
| $E_{Temp}$ | Temporal overlap of the excimer laser pulse with a particle in the particle beam. |
| $E_{Ion}$ | Ionisation efficiency of the excimer laser pulse with respect to the particle composition. |
| $E_{AutoMS}$ | Instrument sampling efficiency in 'Auto Triggering' acquisition mode. The excimer laser is firing at a set repetition rate. |
| $E_{TriggeredMS}$ | Instrument sampling efficiency in 'Second Laser Only' acquisition mode. The excimer laser is fired by a trigger from the second detection stage. |
| $E_{SizedMS}$ | Instrument sampling efficiency in 'Both Lasers' acquisition mode. The instrument is actively sizing and requires a trigger from both detection stages. |

*Acknowledgements.* This work was supported by the UK Natural Environment Research Council (NERC). We
would like to thank Alisdair Macpherson of the Photon Science Institute at the University of Manchester for his
technical support.



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
