# Peer review of "Evaluating the influence of laser wavelength and detection stage geometry on optical detection efficiency in a single particle mass spectrometer."

_Atmospheric Measurement Techniques, 2016_

## Referee Comment (RC1) · Anonymous Referee #1 · 29 Jun 2016

General comments The authors describe the introduction of a custom detection laser system to improve the detection efficiency of a single particle mass spectrometer LAAP-TOF. A numerical model was developed to predict particle detection efficiencies in dependence on various parameters and the results were compared to a data set of ambient measurements during a field campaign. The results of measurements suggest that particle hit efficiency directly correlates to scattering cross section of the particle depending on particle size, particle refractive index and wavelength of incident light.

The manuscript is not very easy to read because the authors use their chosen symbols instead of a written explanations. They expect that readers also have these symbols in mind all time. E.g. L 353: "...particle beam profile would theoretically give E600nmDetect= 0.53..." -> "...particle beam profile would theoretically give an optical detection efficiency of 0.53..."

Some statements are inconsistent: 1. L 427: "Our initial laboratory studies with Detection System A showed that a low detection efficiency in the size range $300 - 800$nm produced a sampling efficiency that was insufficient for the targeted application. However, the hit-rate with this set-up was exceptionally high (close to 1)..." but "A collimated beam maximises the active area of detection whilst minimising the stray light divergence at the light collection optics, but comes at the expense of slightly reduced hit-rates compared to a focussed beam system." (L 476) -> A more precise discussion is necessary and data should be shown, e.g. for "...slightly reduced hit-rates compared to a focussed beam system." (L 478) 2. Is it really an advantage to have higher detection efficiency for spherical particles with diameters between 500 and 800 nm instead of lower efficiency but a wider detection range? 3. What does it help that "The new laser system resulted in an order of magnitude improvement in sensitivity to spherical particles..."? But, in real world applications most of the particles are not spherical! 4. The authors should take into account the laser power density in the focal point in theirs discussions.

-> The presented calculations and data have to be revised and completed.

Specific comments: L 39: "...as well as the the probing of internal mixing state..." -> "...as well as the probing of internal mixing state..."

L 151: "...create a beam with a D4$\sigma$ (second moment width) focal point diameter of $51.2\mu$m..." and Figure 2, caption: "...a 405nm diode based system with a focussed beam (D4$\sigma$ of 50$\mu$m)..." – Which value of focus diameter is correct?

L 207 and L 210: What is meant with "sampling efficiency of the LAAP-TOF"? The manuscript contains discussions about various efficiencies, but it has to be clearly defined here. Probably, the "hit rate", i.e. the ionization efficiency, is meant, but for

which operation mode?

L 209: Please use SI-units: "cc" -> "cm$^3$"

L 212: Please show the parameters for this calculation, e.g. the inlet flow?

L 247: "3.4 Modelling the optical detection geometry" -> "3.4 Modelling the optical detection geometry for collimated detection laser beams"

L260: "The general relationship between particle beam width $\sigma$p and R in Eq. (2) was used to quantify EDetect." -> "The general relationship between particle beam width $\sigma$p and R in Eq. (2) was used to quantify EDetect for a certain particle diameter d."

L 281: "$\sigma$ = 0.10mm" -> "$\sigma$d = 0.10mm"

Figure caption Figure 4 – L 270/274/276/281: Please be consistent using terms, e.g. "$\sigma$d" <-> "$\sigma$detect" <-> "$\sigma$Detect"

Figure 7: is not good visible and too small.

L 345: "The resulting in particle density profile" -> "The resulting particle density profile"

L 348/349: "$\sigma$ = 0.13mm" -> "$\sigma$p = 0.13mm" – Where are the values of $\sigma$p determined, at 1/e2? This has to be mentioned.

L 349: "detection stage 1(120mm from the lens exit), and $\sigma$ = 0.26mm at detection stage 2 (240mm..." compared to Figure caption of Figure 7: "...rear detection stages at a distance of 115mm and 230mm from..." which are the correct values?

Figure caption Figure 7 – L 349: Please be consistent using terms, e.g. "detection stage 1" = "front detection stage"

L 359: What is the value of "$\sigma$d" for this calculation?

L 364/368: "Figure8" -> "Figure 8"

L 374: "cut-off than a the narrow angles" -> "cut-off than the narrow angles"

L 378: "Figure 8B" -> "Figure 8 B"

L 270 and Appendix <-> L 367 and Figure 8: What is the correct name for R? – "active radius of detection" or "effective detection radius" or "active detection radius"

Figure 8: - font sizes of the 4 parts are different. - The indications for the 4 parts (A, B, C, D) are missed. - The indications for the 4 parts (A, B, C, D) should be consistent: NOT "a, b, c, d" in Figure caption - description of Y-axis: "active detection radius" or "active beam diameter"? - "signal-to noise" -> "signal-to-noise" - What is the meaning of System X2 and X10 in Figure 8 C? This should be explained in the text in more detail.

L 349: The authors should use "aerodynamic lens" instead of "lens" if the inlet system is meant and "optical lens" if focusing a laser beam is meant (-> see e.g. L 386) for more clearness.

Figure 8: - D is not discussed in the text. Please add an appropriate discussion. - Why do the authors use a laser power of 300 mW for modelling R at 532 nm? The laser power is originally 1 W. Did they measure the value of 300 mW at the exit of the collimation path?

L 396 to L 400: It is surprising that 250.000 particles are clustered in only three classes. -> Add the clustering conditions of the algorithm. -> Show the patterns of the three classes. -> Add the abundances of the classes as pie chart or table.

Figure 9, - X-axis: "dat" -> "date" - How was the "clustered particle number concentrations" (L 400) calculated? - What is the time resolution of this drawing? - Y-axis: "LAAP-Tof" -> "LAAP-TOF"

Figure 10: - What is the bin width of LAAP-TOF size distributions? - In the drawing "A" and "B" is used in Figure caption "a" and "b". Please correct accordingly. - "RI" is not used in the text. Please use "refractive index"

L 416: "a" -> "at"

L 427 to 429: "Our initial laboratory studies with Detection System A showed that a low detection efficiency in the size range $300 - 800nm$ produced a sampling efficiency that was insufficient for the targeted application." The authors should show their data for this statement.

L 440 to 442: "The particle size distribution of the target application must be considered when choosing the output power and focussing characteristics of the detection laser system if using light collection optics with a narrow collection angle." This is hard to realize for ambient and field measurements because at the beginning of such measurements the size distribution is not known and it can vary during a measurement period. It would be helpful if the authors give a hint how to overcome this problem.

L 445: "the low Eion of some of the silicate mineral dust" – Do the authors have data for this statement? If so, the authors should show it. Usually, mineral particles have high ionization efficiency and mass spectra can be detected down to particle diameters of 200 nm. Therefore, Arizona dust particles are often used for laboratory test measurements.

L 446: "Size dependent detection bias may also influence the sampling efficiency" - Do the authors have data for this statement, e.g. size-resolved offline measurements with an impactor? Otherwise, the statement is speculative and should be deleted.

L 448: "Comparison with the LAAP-TOF size distribution with the APS suggests that the size distributions of the silicate and secondary classes represent the tail of an accumulation mode with a center below 300nm (Figure 10 A?)." – It is not very good visible. Maybe a logarithmic Y-axis gives better evidence. The value of "300 nm" is not shown.

L 503: "evaluation of the effective of particle size and morphology on the overall sampling efficiency"?

L 506: "ESizedMS could be improved by reducing the length of the particle flight path by

shortening the analyser housing." - The authors should propose possible instrumental improvements.

References: References are sometimes incomplete and have to be revised substantially. The format should match the rules of the journal! e.g.: - Murphy, D. M.: Guest Editor : Albert Viggiano THE DESIGN OF SINGLE PARTICLE LASER MASS SPECTROMETERS, pp. 150–165, doi:10.1002/mas, 2007 -> doi:10.1002/mas.20113 and/or Mass Spectrometry Reviews, 2007, 26, 150– 165. -> reference of "guest editor" is unusual.

- Jonsson, H. H., Wilson, H. C., and Brock, R. G.: Performance of a focused cavity aerosol spectrometer for measurements in the stratosphere of particle size in the 0.06-2.0 um diamater range., American Meteorological Society, 1995. -> H.H. Jonsson, J.C. Wilson, C.A. Brock, R.G. Knollenberg, T.R. Newton, J.E. Dye, D. Baumgardner, S. Borrmann, G.V. Ferry, R. Pueschel, Dave C. Woods, and Mike C. Pitts: Performance of a Focused Cavity Aerosol Spectrometer for Measurements in the Stratosphere of Particle Size in the 0.06–2.0-$\mu$m-Diameter Range. Journal of Atmospheric and Oceanic Technology, 1995, 12:1, 115-129.
* * *

---

## Referee Comment (RC2) · Anonymous Referee #2 · 12 Jul 2016

Review 2

Atmos. Meas. Tech. Discuss., doi:10.5194/amt-2016-150, 2016 Manuscript under review for journal Atmos. Meas. Tech. Published: 12 May 2016

Evaluating the influence of laser wavelength and detection stage geometry on optical detection efficiency in a single particle mass spectrometer.

The scientific approach is sound and the contribution important. However, the paper is really not easy to read, it is indeed a technical paper, but I think the flow should be improved. After major reorganization, it surely can be accepted for publication.

Reference list. I found the introduction and the overall reference list very poor. Given

the >20 years of SPMS, I would have hoped for a more sound reference review. Papers like Prather and Moffet (PNAS 2009) should be included. I would make a short review of ATOFMS-SPMS studies (San Diego, Birmingham, recent Canada-Ireland work, Germany) on detection efficiencies and particle matrix effect and so on. It would definitely help this paper.

The paper needs some bullets point or a better flow. Page 4 line 110-120, page 6 line 140-170, section 3.3 for example.

Figure 2. Perhaps explain better the original setting and the modified settings, and the consequences?

I am not sure I follow the result section, especially section 4.1 and 4.2. Perhaps a paragraph introducing the results and the sections?

Figure 11. Is it appropriate at the end of the discussion?

In summary, I am convinced of the new improvement of the instrument and the results are sound. I think they can be better presented, both in the introduction (state of the art of SPMS), and a better organization of the scattered results difficult to follow. Scientific context is sound and accepted, a better presentation of the results is needed. The paper at this stage is very difficult to follow and is not the easiest read.

---

## Author Comment (AC1) · 31 Aug 2016

*Evaluating the influence of laser wavelength and detection stage geometry on optical detection efficiency in a single particle mass spectrometer*

Dear Anonymous Referee #1,

We sincerely appreciate the detail with which you have reviewed our manuscript and the constructive comments given. Our response to your comments is given below.

On behalf of the authors,

Nick Marsden

The response to the review is structured as follows: The original reviewer comments are given in black, followed by the author response in blue font colour. Changes that will be made to the manuscript are then indicated in red.

Authors reply to review comments by Anonymous Referee #1:

General comments: The authors describe the introduction of a custom detection laser system to improve the detection efficiency of a single particle mass spectrometer LAAP-TOF. A numerical model was developed to predict particle detection efficiencies in dependence on various parameters and the results were compared to a data set of ambient measurements during a field campaign. The results of measurements suggest that particle hit efficiency directly correlates to scattering cross section of the particle depending on particle size, particle refractive index and wavelength of incident light.

The manuscript is not very easy to read because the authors use their chosen symbols instead of a written explanations. They expect that readers also have these symbols in mind all time. E.g. L 353: "particle beam profile would theoretically give $E_{600nmDetect} = 0.53$" -> "particle beam profile would theoretically give an optical detection efficiency of 0.53"

Symbols are needed for the parameterisation of the model and have to be defined accurately in the text. Symbols follow the convention used in other publications that deal with particle beam geometry e.g. Huffman 2005. A list of the symbols is provided in Appendix A. However, we acknowledge that this makes the document hard to read.

The full written explanation will be used when a parameter is used for the first time in each section and subsection so that the meaning of the symbol is refreshed to the reader.

Some statements are inconsistent: 1. L 427: "Our initial laboratory studies with Detection System A showed that a low detection efficiency in the size range 300− 800nm produced a sampling efficiency that was insufficient for the targeted application. However, the hit-rate with this set-up was exceptionally high (close to 1)" …but "A collimated beam maximises the active area of detection whilst minimising the stray light divergence at the light collection optics, but comes at the expense of slightly reduced hit-rates compared to a focussed beam system."

There are numerous definitions of the term hit-rate in the literature. The definition of hit-rate in this paper is given (page 8 L224);

"The probability of hitting a particle and obtaining a mass spectrum within this 2DG-PDF is known as the Hit-Rate efficiency ($E^d_{Hit}$), and is the product of the geometrical overlap of the excimer laser focal point with the particle beam ($E^d_{Geom}$), the temporal overlap of the peak UV laser power with the presence of a particle in ionisation region ($E^d_{Temp}$), and the ionisation efficiency of the UV laser ($E^d_{Ion}$) with respect to the particle composition".

It is therefore possible to have high detection efficiency and a low hit-rate if a particle is detected in the optical detection stage but no ions are produced in the ion source.

The meaning of hit rate will be clarified early in the document where the term is first use (page 1 L42).

(L 476) -> A more precise discussion is necessary and data should be shown, e.g. for "slightly reduced hit-rates compared to a focussed beam system."

See the definition of hit-rate above. The data is shown in figure 6 lower panel. We acknowledge that this is not clearly described.

The caption for figure 6 will be improved so that is clear which parameters the plot represents. The full written explanation of $E_{Detect}$ will be given in the caption. The hit-rate calculation will be specifically described in the text at (L334-342). 'Hit rate = Number of particle that produce a mass spectra / number of particles that are detected at the optical detection stage'.

(L 478) 2. Is it really an advantage to have higher detection efficiency for spherical particles with diameters between 500 and 800 nm instead of lower efficiency but a wider detection range?

Yes. The application is for the measurement of course mode dust particles at relatively low number concentrations as stated in Section3.2. The new detection system may actually have a wider continuous detection range with some particle types because R=0 for the 405nm diode system for particles ~1μm and 2μm in diameter.

3. What does it help that "The new laser system resulted in an order of magnitude improvement in sensitivity to spherical particles"? But, in real world applications most of the particles are not spherical!

There will be an improvement for all particles shapes, but only spherical particle were measured because they are easier to model. Detection System A and Detection System B will both be affected by the difference in scattering phase function of spherical and non-spherical particles. In addition, non-spherical particles will produce a more divergent particle beam. The effective detection radius will be more important for larger particle beam widths as shown in Figure 3.

The implications non-spherical particles will be discussion with the ambient data in Section 4.3 L456, with reference to the scattering phase function and the particle beam divergence.

4. The authors should take into account the laser power density in the focal point in theirs discussions.

The model considers the intersection of the particle beam and detection beam as a cross-sectional area, hence we construct the model with power intensity in 2D space.

-> The presented calculations and data have to be revised and completed.
Specific comments:

L 39: "...as well as the the probing of internal mixing state..." ->"...as well as the probing of internal mixing state..."
Will be corrected as described
L 151: "...create a beam with a D4σ (second moment width) focal point diameter of 51.µm.." and Figure 2, caption: "...a 405nm diode based system with a focussed beam (D4σ of 50µm)..." – Which value of focus diameter is correct?

The value 51.2µm will be used in all instances.

L 207 and L 210: What is meant with "sampling efficiency of the LAAP-TOF"? The manuscript contains discussions about various efficiencies, but it has to be clearly defined here. Probably, the "hit rate", i.e. the ionization efficiency, is meant, but for which operation mode?

This sentence will be removed

L 209: Please use SI-units: "cc" -> "cm$_3$"

Will be corrected as described

L 212: Please show the parameters for this calculation, e.g. the inlet flow?

Inlet flow will be added and the paragraph re-written.
'Previous studies at this location have recorded a multi-modal aerosol size distribution with number concentrations in the size range 0:5□2:5m of between 10-100 cm3 for clean maritime conditions and dust events respectively (Kandler et al., 2011).With an inlet flow rate of 0.078 L/min, the LAAP-TOF was required to accumulate mass spectra for 1-10% of particle present during the ICE-D campaign in order to measure sized resolved composition at a temporal resolution of several particles per minute'.

L 247: "3.4 Modelling the optical detection geometry" -> "3.4 Modelling the optical detection geometry for collimated detection laser beams"

Will be corrected as described

L260: "The general relationship between particle beam width σp and R in Eq. (2) was used to quantify EDetect." -> "The general relationship between particle beam width σ p and R in Eq. (2) was used to quantify EDetect for a certain particle diameter d."

Will be corrected as described

L 281: "σ= 0.10mm" -> "σd = 0.10mm" Figure caption Figure 4 – L 270/274/276/281: Please be consistent using terms, e.g."σd" <-> "σdetect" <-> "σDetect" Figure 7: is not good visible and too small.

σ$_d$ will be used in all instances. Figure 7 will be made visible.

L 345: "The resulting in particle density profile" -> "The resulting particle density profile"

Will be corrected as described

L 348/349: "σ= 0.13mm" -> "σp = 0.13mm" – Where are the values of σp determined, at 1/e2? This has to be mentioned.

This is the width from the Gaussian curve fit to the particle beam density distribution in figure7.

The symbol/subscript will be updated to $\sigma_d$.

L 349: "detection stage 1(120mm from the lens exit), and σ= 0.26mm at detection stage 2 (240mm..." compared to Figure caption of Figure 7: "...rear detection stages at a distance of 115mm and 230mm from.." which are the correct values?

The distances 125mm and 240mm will be used in all instances respectively.

Figure caption Figure 7 – L 349: Please be consistent using terms, e.g. "detection stage 1" = "front detection stage"

Detection Stage 1 and Detection Stage 2 will be used in all instances.

L 359: What is the value of "σd" for this calculation?

The value of $\sigma_d$ = 0.0825mm will be added to the sentence.

L 364/368: "Figure8" -> "Figure 8"

Will be corrected as described.

L 374: "cut-off than a the narrow angles" -> "cut-off than the narrow angles"
corrected

L 378: "Figure 8B" -> "Figure 8 B"

Will be corrected as described.

L 270 and Appendix <-> L 367 and Figure 8: What is the correct name for R? – "active radius of detection" or "effective detection radius" or "active detection radius"

The name for R will be Effective Detection Radius in all instances.

Figure 8: - font sizes of the 4 parts are different. - The indications for the 4 parts (A, B, C, D) are missed. - The indications for the 4 parts (A, B, C, D) should be consistent: NOT "a, b, c, d" in Figure caption - description of Y-axis: "active detection radius" or "active beam diameter"? - "signal-to noise" -> "signal-to-noise" - What is the meaning of System X2 and X10 in Figure 8 C? This should be explained in the text in more detail.

Figure 8 will be remade to the following graphic. The caption updated to give a definition of all abbreviations. Part D will be removed from this figure.

L 349: The authors should use "aerodynamic lens" instead of "lens" if the inlet system is meant and "optical lens" if focusing a laser beam is meant (-> see e.g. L 386) for more clearness.

Will be corrected as described.

Figure 8: - D is not discussed in the text. Please add an appropriate discussion. - Why do the authors use a laser power of 300 mW for modelling R at 532 nm? The laser power is originally 1 W. Did they measure the value of 300 mW at the exit of the collimation path?

300mW was indeed measured at the exit of the collimation path.
Part D will be removed from Figure 8. The effect of Refractive index will be considered in figure 10 only. Part C will be added to Figure 10 that shows R for particles with a refractive index of 1.33. The actual laser power delivered to the optical detection stage will be stated in the instrument description (Section 2 L156).

L 396 to L 400: It is surprising that 250.000 particles are clustered in only three classes. -> Add the clustering conditions of the algorithm. -> Show the patterns of the three classes. -> Add the abundances of the classes as pie chart or table.

A detailed analysis of the ICE-D data set will be presented in a future publication. For the purposes of this manuscript we are interested in the particle counting statistics for the main particle composition types. The data was processed with 6 clusters and then manually reduced to 3 on inspection. 4 classes of particles were dominated by NaCl markers and were reduced to 1 class. Shot -to-shot variation in the mass calibration are

responsible for some of cluster centres. The discussion of the origins of the classes is complex and beyond the scope of this document.

The cluster conditions will be added to the ambient data result section L396-400. A pie chart showing the abundances of the classes will be added to figure9. The spectral patterns of the classes will be put into appendix C so that the document remains focussed on the particle counting efficiency.

Figure 9, - X-axis: "dat" -> "date" - How was the "clustered particle number concentrations" (L 400) calculated? - What is the time resolution of this drawing? - Y-axis: "LAAP-Tof" -> "LAAP-TOF"

Clustered particle number concentrations are reported in the OEM software for cluster analysis.

Line 400 will read: 'A 7 day time series of the clustered particle number concentrations reported by the LAAP-TOF Data Analysis Software after cluster analysis is shown in Figure 9'

Axis labels will be corrected in Figure 9.

Figure 10: - What is the bin width of LAAP-TOF size distributions? - In the drawing "A" and "B" is used in Figure caption "a" and "b". Please correct accordingly. - "RI" is not used in the text. Please use "refractive index"

Will be corrected as described.

L 416: "a" -> "at"

Will be corrected as described.

L 427 to 429: "Our initial laboratory studies with Detection System A showed that a low detection efficiency in the size range 300−800nm produced a sampling efficiency that was insufficient for the targeted application." The authors should show their data for this statement.

The data this referring to is the instrument performance data for Detection System A given in Figure 6. The sampling efficiency required for the target application is discussed at L208.

This sentence will be re-written as:
"Our measurement of instrument performance with Detection System A showed a low detection efficiency in the size range 300−800nm (Figure 6) resulting in a sampling efficiency that was insufficient for the temporal resolution required for the targeted application."

L 440 to 442: "The particle size distribution of the target application must be considered when choosing the output power and focussing characteristics of the detection laser system if using light collection optics with a narrow collection angle." This is hard to realize for ambient and field measurements because at the beginning of such measurements the size distribution is not known and it can vary during a measurement period. It would be helpful if the authors give a hint how to overcome this problem.

The size dependent detection bias may be mitigated by; reducing the particle beam at the detection stage by shortening the analyser housing, increasing the solid angle at which light is collect e.g. elliptical mirror, or a using mixed wavelength solution described later in the text at L464.

We will add an extra paragraph at L457 that describes how shortening the analyser and using elliptical mirrors would mitigate this problem with reference to the design of the PALMS (short housing) and the ATOFMS, SPLAT and ALABAMA instruments (elliptical mirror)

L464 will be re-written as:
'A detection laser consisting of mixed wavelengths may help mitigate the size dependent detection bias evident with the near-forward light collection system utilised in LAAP-TOF'.

L 445: "the low Eion of some of the silicate mineral dust" – Do the authors have data for this statement? If so, the authors should show it. Usually, mineral particles have high ionization efficiency and mass spectra can be detected down to particle diameters of 200 nm. Therefore, Arizona dust particles are often used for laboratory test measurements.

This will be changed to:
'differences in the ionisation efficiency of ambient aerosols compared to laboratory generated PSL.'

L 446: "Size dependent detection bias may also influence the sampling efficiency" - Do the authors have data for this statement, e.g. size-resolved offline measurements with an impactor? Otherwise, the statement is speculative and should be deleted.

We believe that the comparison of the model with the ambient data on Figure 10 shows this.

L446 will be changed to:
'The size dependent detection bias predicted by the model of the optical detection geometry may also influence the sampling efficiency '

L 448: "Comparison with the LAAP-TOF size distribution with the APS suggests that the size distributions of the silicate and secondary classes represent the tail of an accumulation mode with a centre below 300nm (Figure 10 A?)." – It is not very good visible. Maybe a logarithmic Y-axis gives better evidence. The value of "300 nm" is not shown.

Figure 10 A will be made good visible with a logarithmic scale.

[Figure]

Line 448 will be re-written as:
'Comparison with the LAAP-TOF size distribution with the APS suggests that
the multi-modal size distributions of the silicate and secondary classes maybe the result
of the optical detection bias rather than a reflection of the true particle size distribution.
(Figure 10 A)'

L 503: "evaluation of the effective of particle size and morphology on the overall sampling efficiency"?

L503 will be changed to:
"evaluation of the effect of particle size and morphology on the overall sampling efficiency"

L 506: "ESizedMS could be improved by reducing the length of the particle flight path by shortening the analyser housing." - The authors should propose possible instrumental improvements.

The multi-wavelength detection system, shortening the instrument housing and increasing the angle of light collection are the suggested improvements.

The final paragraph of the conclusions will be updated so that the suggested improvements appear in the same sentence.

References: References are sometimes incomplete and have to be revised substantially. The format should match the rules of the journal! e.g.: - Murphy, D. M.: Guest Editor : Albert Viggiano THE DESIGN OF SINGLE PARTICLE LASER MASS SPECTROMETERS, pp. 150–165, doi:10.1002/mas, 2007 -> doi:10.1002/mas.20113 and/or Mass Spectrometry Reviews, 2007, 26, 150– 165. -> reference of "guest editor" is

unusual.

- Jonsson, H. H., Wilson, H. C., and Brock, R. G.: Performance of a focused cavity aerosol spectrometer for measurements in the stratosphere of particle size in the 0.06-2.0 um diamater range., American Meteorological Society, 1995. -> H.H. Jonsson, J.C. Wilson, C.A. Brock, R.G. Knollenberg, T.R. Newton, J.E. Dye, D. Baumgardner, S. Borrmann, G.V. Ferry, R. Pueschel, Dave C. Woods, and Mike C. Pitts: Performance of a Focused Cavity Aerosol Spectrometer for Measurements in the Stratosphere of Particle Size in the 0.06–2.0-

μ

m-Diameter Range. Journal of Atmospheric and Oceanic

Technology, 1995, 12:1, 115-129.

All references will be reviewed and corrected.

---

## Author Comment (AC2) · 31 Aug 2016

*Evaluating the influence of laser wavelength and detection stage geometry on optical detection efficiency in a single particle mass spectrometer*

Dear Anonymous Referee #2,

Many thanks for the your suggestion on how to improve the flow and readability of out manuscript . Our response to your comments is given below.

On behalf of the authors,

Nick Marsden

The response to the review is structured as follows: The original reviewer comments are given in black, followed by the author response in blue font colour. Changes that will be made to the manuscript are then indicated in red.

Authors reply to review comments by Anonymous Referee #2:

The scientific approach is sound and the contribution important. However, the paper is really not easy to read, it is indeed a technical paper, but I think the flow should be improved. After major reorganization, it surely can be accepted for publication.

We acknowledge that the paper is difficult to read and will make changes to improve the readability and flow of the manuscript.

Reference list. I found the introduction and the overall reference list very poor. Given the >20 years of SPMS, I would have hoped for a more sound reference review. Papers like Prather and Moffet (PNAS 2009) should be included. I would make a short review of ATOFMS-SPMS studies (San Diego, Birmingham, recent Canada-Ireland work, Germany) on detection efficiencies and particle matrix effect and so on. It would definitely help this paper.

We accept that more referencing is needed in the introduction.

Reference to the direct measurement of the scattering cross section with optical detection in SPMS (Prather and Moffet 2009) will be made (page 2 L53).

New paragraph to be added (page 3 L79) that gives an overview of the qualitative nature of SPMS and how variation of instrument function affect particle counting statistics.

New paragraph to be added at (page 3 L79) that gives a brief overview of recent ATOFMS publications as examples of the type of application for SPMS.

The paper needs some bullets point or a better flow. Page 4 line 110-120, page 6 line140-170, section 3.3 for example.

Instrument acquisition modes (Page 4  L110-120) will be written as bullet points.

The detailed descriptions of the build of optical detection systems (Page 6 L148-170) will be moved to an appendix to improve the flow. A table will be added (Page 6 L148) that summarises the key differences between the optical detection systems.

Explanation of the parameters in section 3.3 requires a narrative that does not lend itself to bullet points.

Title to Section 3.3 to be changed to 'Instrument Performance Definitions'

Figure 2. Perhaps explain better the original setting and the modified settings, and the consequences?

Text at page 6 L140 will be changed to concentrate on the important differences consequences of the two detection systems, with a table for easy reference.

I am not sure I follow the result section, especially section 4.1 and 4.2. Perhaps a paragraph introducing the results and the sections?

A new introduction to the Results section will be written. The results will be organised into a new set of subsections.

4.1 Instrument Performance Measurements
4.2 Particle Beam Width Measurements
4.3 Derivation of the Transfer Function
4.4 Model predictions of the effective detection radius with selected wavelengths, collection angles and signal-to-noise conditions.
4.5 Ambient Measurements

Figure 11. Is it appropriate at the end of the discussion?

This figure is referred to as part of the discussion of possible solutions to the particle detection bias.

Figure 11 will be moved to the middle of the discussion section.

In summary, I am convinced of the new improvement of the instrument and the results are sound. I think they can be better presented, both in the introduction (state of the art of SPMS), and a better organization of the scattered results difficult to follow. Scientific context is sound and accepted, a better presentation of the results is needed.

The paper at this stage is very difficult to follow and is not the easiest read.